# Baby skyrmions in Chern ferromagnets and topological mechanism for spin-polaron formation in twisted bilayer graphene

Eslam Khalaf [1,2] ✉ & Ashvin Vishwanath[1]

The advent of moiré materials has galvanized interest in the nature of charge carriers in topological bands. In contrast to conventional materials with electron-like charge carriers, topological bands allow for more exotic possibilities where charge is carried by nontrivial topological textures, such as skyrmions. However, the real-space description of skyrmions is ill-suited to address the limit of small skyrmions and to account for momentum-space band features. Here, we develop a momentum-space approach to study the formation of the smallest skyrmions – spin polarons, formed as bound states of an electron and a spin flip – in topological ferromagnets. We show that, quite generally, there is an attraction between an electron and a spin flip that is purely topological in origin, promoting the formation of spin polarons. Applying our results to twisted bilayer graphene, we identify a range of parameters where spin polarons are formed and discuss their possible experimental signatures.

There has been much recent interest in narrow Chern bands that spontaneously develop ferromagnetic order, following their appearance in a variety of Moiré materials[1–8]. The bands of magic-angle twisted bilayer graphene (MATBG) can, for example, be viewed as complementary Chern bands residing on opposite sublattices[9–12], where spin, valley, and sublattice polarization leads to Chern insulators as observed in experiment[5,13]. The key question is, what is the nature of charge carriers associated with doping these generalized Chern ferromagnets? The answer will have implications for the entire phase diagram of various Moiré materials, and could hold the key to explaining mysteries such as the doping dependence and origin of superconductivity in MATBG and related structures. The simplest example of a Chern band, Landau levels, have previously been shown to exhibit a ferromagnetic ground state at unit filling[14,15]. Despite the simplicity of the ground state, charge excitations can be very nontrivial. In addition to single electron quasiparticles corresponding to adding an electron with a reversed spin, this system also hosts charged skyrmions—smooth textures of the ferromagnetic order that carry an electric charge proportional to their topological winding[14,16]. The smallest nontrivial limit of a skyrmion is a quasiparticle bound to a single-spin flip, i.e., a spin-polaron. (For a discussion of spin-polarons in non-topological ferromagnets, see e.g., ref. 17).

Upon passing from Landau levels to Chern bands, questions about nontrivial charge carriers become more subtle. On the one hand, these bands share the same topological character as Landau level and are thus expected to realize similar charge excitations. On the other hand, in comparison to Landau levels, Chern bands possess a rich reciprocal space structure, including band dispersion and variation of band geometric features such as Berry curvature over the Brillouin Zone. These features are not readily incorporated in the standard way of treating skyrmion excitations as real space textures. Furthermore, although skyrmions can be smoothly shrunk to electrons, the standard description of the two excitations could not be more different: the former is a real space texture whose energy is computed using effective field theory[14,15,18–21] or real space variational methods[22–24] while the latter is a momentum eigenstate whose energy is obtained from momentum space Hartree–Fock. One route to connecting real and momentum space descriptions begins with small skyrmions, and in particular spin polarons, which

[1]Department of Physics, Harvard University, Cambridge, MA 02138, USA. [2]Department of Physics, The University of Texas at Austin, Austin, TX 78712, USA. ✉e-mail: eslam.khalaf@austin.utexas.edu

are more naturally described as electrons dressed by spin flips[25] rather than as real space textures. The former picture furnishes a momentum space approach that can easily incorporate the reciprocal space properties of Chern insulators and can also facilitate the use of powerful diagrammatic techniques. However, even at the outset, this program poses a puzzle: how is information about band topology, necessary for electrically charged skyrmions, incorporated in the dressed electron picture, i.e., how does a localized excitation in momentum space detect the topology of the entire band?

In this work, we take the first step towards a momentum space characterization of nontrivial charge excitations in Chern ferromagnets by studying the formation of the smallest skyrmions, the spin polaron, consisting of an electron dressed by one spin flip. We show that the matrix elements of an arbitrary density–density interaction $V_{\mathbf{q}}$ between an electron-magnon state with magnon momentum $\mathbf{q}$ and one with momentum $\mathbf{q}'$ is $\propto i\mathbf{q} \wedge \mathbf{q}' \frac{2\pi C}{A_{\mathrm{BZ}}} V_{\mathbf{q}-\mathbf{q}'}$ at small $\mathbf{q}$ and $\mathbf{q}'$, where $C$ is the Chern number. This result can be understood by rewriting the interaction as $\mathbf{d} \cdot \mathbf{E}$ where the electric field $\mathbf{E}$ is given by $\mathbf{E} = -\mathbf{q}V_{\mathbf{q}}$ and the magnon dipole moment $\mathbf{d}$ given by $\mathbf{d} = \frac{2\pi C}{A_{\mathrm{BZ}}} e \wedge \mathbf{q}'$ is a consequence of the relationship between momentum and dipole moment in a chern band[15,26]. Remarkably, this implies there is an attractive interaction between an electron and a spin flip for any repulsive interaction $V$, which takes place in the $(p_x + ip_y)$-wave channel, as a consequence of the band topology.

We further investigate the conditions under which such attractive interaction leads to the formation of a bound state and apply the results to models of twisted bilayer graphene. Although the spectrum of single-particle excitations in such systems has been obtained from self-consistent Hartree–Fock studies[11,12,27,28] whose results are exact in certain limits[29,30], the existence of other low-lying charged excitations, e.g. skyrmions or spin-polarons, implies that a single-quasiparticle based description is insufficient to capture the physics on doping correlated insulators. Such unconventional excitations were proposed by the authors and co-workers to play a crucial role in the "skyrmion mechanism" of superconductivity[20,21].

To solve this problem, we exploit the fact that the Hilbert space of an electron + a single-spin flip only scales as $N^2$ for a system with $N$ unit cells which allows us to solve this problem for relatively large system sizes. Our results can be summarized as follows: (i) In the limit of vanishing quasiparticle dispersion (i.e., the non-interacting dispersion + the interaction-generated dispersion), we find that the spin polaron is always lower in energy than the electron, with its energy increasing as the Berry curvature gets more concentrated. (ii) The existence of the spin polaron as a bound state is very sensitive to band topology and it is lost if we drive a phase transition to Chern trivial bands. (iii) We find that there is a critical value for the effective mass of the quasiparticle bands, beyond which the single electron is lower in energy than an electron dressed with a spin flip. In this limit, although the spin polaron does not exist as a stable bound state, it can still influence physics as a resonance. Our results serve as a bridge between momentum space single-particle excitations and real space skyrmion excitations, with the energy of the spin polarons computed here providing a strict upper bound on the energy of skyrmion excitations. Furthermore, our results show that a description in terms of single-particle excitations, even when they are "exact", is generally incomplete to understand the physics of charge doping in a Chern-ferromagnet. In the end, we discuss the implications of these results for the phenomenology of TBG.

## Results
### General formalism
We consider the Hamiltonian of a density–density interaction $V_{\mathbf{q}}$ projected onto a pair of SU(2)-symmetric bands with single-particle dispersion $\epsilon_0(\mathbf{k})$ and wavefunctions $|u_{\mathbf{k}}\rangle$:

$$\mathcal{H} = \sum_{\mathbf{k},\sigma=\uparrow,\downarrow} c^{\dagger}_{\mathbf{k},\sigma}\epsilon_0(\mathbf{k})c_{\mathbf{k},\sigma} + \frac{1}{2A}\sum_{\mathbf{q}} V_{\mathbf{q}}\delta\rho_{\mathbf{q}}\delta\rho_{-\mathbf{q}}, \quad (1)$$

where $\delta\rho_{\mathbf{q}} = \rho_{\mathbf{q}} - \bar{\rho}_{\mathbf{q}}$ is the projected density measured relative to a certain reference chosen such that the interacting piece of the Hamiltonian annihilates the ferromagnetic state at half-filling (see ref. 12 for details). The projected density operator is given by

$$\rho_{\mathbf{q}} = \sum_{\mathbf{k}} c^{\dagger}_{\sigma,\mathbf{k}}c_{\sigma,\mathbf{k}+\mathbf{q}}\lambda_{\mathbf{q}}(\mathbf{k}), \qquad \lambda_{\mathbf{q}}(\mathbf{k}) = \langle u_{\mathbf{k}}|u_{\mathbf{k}+\mathbf{q}}\rangle \quad (2)$$

If the bare dispersion $\epsilon_0$ is sufficiently small, the ground state of the Hamiltonian (1) is a ferromagnet $|\downarrow\rangle = \prod_{\mathbf{k}} c^{\dagger}_{\downarrow,\mathbf{k}}|0\rangle$ (annihilated by $\delta\rho_{\mathbf{q}}$ for all $\mathbf{q}$), with total spin $S = \frac{N}{2}$ which we choose to have $S_z = -\frac{N}{2}$. Single-particle excitations with charge $\mp e$ are given by $|\mathbf{k}\rangle_e = c^{\dagger}_{\uparrow,\mathbf{k}}|\downarrow\rangle$ and $|\mathbf{k}\rangle_h = c_{\downarrow,\mathbf{k}}|\downarrow\rangle$, respectively. The state $|\mathbf{k}\rangle_{e/h}$ has $S_z = -\frac{N-1}{2}$ and total spin $S = \frac{N-1}{2}$ and its energy is given exactly (up to an irrelevant constant) by $\mathcal{H}|\mathbf{k}\rangle_{e/h} = \epsilon_{e/h}(\mathbf{k})|\mathbf{k}\rangle_{e/h}$. The quasiparticle dispersion $\epsilon_{e/h}(\mathbf{k})$ is given by $\epsilon_{e/h}(\mathbf{k}) = \pm\epsilon_0(\mathbf{k}) + \epsilon_F(\mathbf{k})$ where the interaction-generated dispersion $\epsilon_F(\mathbf{k}) = \frac{1}{2A}\sum_{\mathbf{q}} V_{\mathbf{q}}|\lambda_{\mathbf{q}}(\mathbf{k})|^2$ is nothing but the Fock energy, which gives rise to a nontrivial band dispersion whenever the magnitude of the form factor $|\lambda_{\mathbf{q}}(\mathbf{k})|$ is $\mathbf{k}$-dependent. In the following, we will mainly focus on the electron bands and drop the $e$ subscript.

We now consider a basis of states containing an electron and a spin flip which is obtained from the ground state ferromagnet by creating two electrons with spin up and a hole with spin down

$$|\mathbf{k}_{e1}, \mathbf{k}_{e2}, \mathbf{k}_h\rangle = c^{\dagger}_{\uparrow,\mathbf{k}_{e1}} c^{\dagger}_{\uparrow,\mathbf{k}_{e2}} c_{\downarrow,\mathbf{k}_h}|\downarrow\rangle \quad (3)$$

The effective Hamiltonian in the two-electron/one-hole sector is defined as $H^{2e1h}_{\mathbf{k}'_{e1},\mathbf{k}'_{e2},\mathbf{k}'_h;\mathbf{k}_{e1},\mathbf{k}_{e2},\mathbf{k}_h} = \langle \mathbf{k}'_{e1}, \mathbf{k}'_{e2}, \mathbf{k}'_h|\mathcal{H}|\mathbf{k}_{e1}, \mathbf{k}_{e2}, \mathbf{k}_h\rangle$ whose explicit form is provided in the methods section. The Hamiltonian $H^{2e1h}$ acts on the state $|\mathbf{k}_{e1}, \mathbf{k}_{e2}, \mathbf{k}_h\rangle$ by shifting two of the three momenta $\mathbf{k}_{e1}, \mathbf{k}_{e2}$ and $\mathbf{k}_h$ such that the total momentum $\mathbf{k} = \mathbf{k}_{e1} + \mathbf{k}_{e2} - \mathbf{k}_h$ is conserved. Thus, we can separately diagonalize the Hamiltonian for each total momentum sector labeling the states by the two electronic momenta $\mathbf{k}_{e1}$ and $\mathbf{k}_{e2}$. Due to fermionic anticommutation relations, these only label $\frac{N(N-1)}{2}$ distinct states for a grid with $N$ points.

Notice that the Hilbert space spanned by the states (3) corresponds to states with definite $S_z = -\frac{N-3}{2}$ but with total spin $S = \frac{N-3}{2}$ or $\frac{N-1}{2}$. Since the Hamiltonian (1) conserves the total spin, we can label the eigenstates of $H^{2e1h}$ by $S = \frac{N-1}{2}, \frac{N-3}{2}$. The explicit form of the total spin operator in the basis (3) is provided in the methods section. Notice that the Hamiltonian $H^{2e1h}$ always has an eigenstate with total spin $S = \frac{N-1}{2}$ and energy $\epsilon(\mathbf{k})$ generated by acting with the spin raising operator, which commutes with $\mathcal{H}$, on the single-particle excitation $|\mathbf{k}\rangle$.

Our goal is to understand the energy competition between single-particle excitations ($S = \frac{N-1}{2}$) and those dressed by a spin flip ($S = \frac{N-3}{2}$). If the latter is lower in energy, this indicates the existence of a bound state of an electron and spin flip, a spin polaron. So far, our discussion has been very general. Our model has as inputs the interaction $V_{\mathbf{q}}$, the bare dispersion $\epsilon_0(\mathbf{k})$ and the form factors $\lambda_{\mathbf{q}}(\mathbf{k})$. It is worth mentioning that usually, $\epsilon_0(\mathbf{k})$ and $\lambda_{\mathbf{q}}(\mathbf{k})$ are generated from the same microscopic model, so they are not always independent. However, we will choose to think of them as independent inputs, which allows us to take into account the effect of remote bands on dispersion. To study the competition between single-particle excitations and spin-dressed excitations, we use a class of continuum models that continuously interpolates between the LLL and the narrow Chern bands of TBG. This class of models provides an excellent playground to study the formation of spin polarons by allowing independent tuning of bandwidth, band topology, and band geometry.

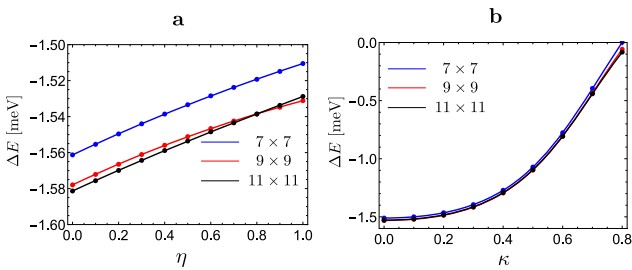

**Fig. 1 | Binding energy of the polaron-bound state for flat quasiparticle dispersion.** The plot of the binding energy for flat quasiparticle dispersion (**a**) as we extrapolate between the LLL in the uniform field and chiral TBG wavefunctions, **b** as

a function of the chiral ratio $\kappa$, **c** as we tune the Chern number of the band by changing the bottom layer sublattice potential $\delta_{\text{bottom}}$ for fixed top layer sublattice potential $\delta_{\text{top}} = 10$ meV, and **d** as a function of the screening gate distance.

## Twisted bilayer graphene bands

Before discussing our results, it is instructive to briefly review the continuum model for twisted bilayer graphene (TBG)[31–33], which consists of two Dirac Hamiltonians coupled through a Moiré potential. The latter has a matrix structure in the sublattice space and can be parametrized by two hopping parameters for intra- and inter-sublattice tunneling, denoted by $w_{\text{AA}}$ and $w_{\text{AB}}$ whose ratio has been estimated to be around $\kappa = \frac{w_{\text{AA}}}{w_{\text{AB}}} \approx 0.5 - 0.8$[34–36]. A particularly interesting limit called the chiral limit corresponds to the case $\kappa = 0$[9]. The wavefunctions of the model in this limit are sublattice-polarized with Chern number $\pm 1$ and has been shown to be equivalent to the lowest Landau level of a Dirac particle in an inhomogeneous periodic magnetic field[37] providing a direct relation between this model and Landau level physics. Even away from the $\kappa = 0$ limits, we can define such a sublattice-polarized basis where the bands have well-defined Chern numbers[12]. This leads to a total of 4 + 4 bands with Chern numbers +1 and −1.

The results of this work apply to TBG under two assumptions. First, we employ the independent Chern sector approximation by neglecting the coupling between Chern sectors. This approximation retains the U(4) symmetry rotating the bands within each Chern sector by ignoring inter-Chern dispersion and inter-Chern wavefunction overlaps. These are both relatively small perturbations of the U(4) × U(4) model that has comparable magnitude[12]. This approximation implies separate spin conservation in each sector that allows us to sharply distinguish the electron from the polaron (otherwise, the two can, in principle, tunnel into each other). Second, we assume only two active bands with SU(2) spin rotation. The remaining bands are assumed to be completely filled or empty and only influence the problem by changing the dispersion $\epsilon_0(\mathbf{k})$ through Hartree corrections, as we will discuss later. In the end, we will discuss the validity of these assumptions.

## Flat quasiparticle dispersion

We will start by focusing on the limit of flat quasiparticle dispersion where the single-particle dispersion and the interaction-generated dispersion exactly cancel for the electron band, $\epsilon_0(\mathbf{k}) = -\epsilon_F(\mathbf{k})$. It is important to emphasize here that this limit is distinct from the flat band limit where the bare dispersion vanishes $\epsilon_0(\mathbf{k}) = 0$ that occurs for the chiral model at the magic-angle, assuming there are no other sources of dispersion. In particular, we will show later that this limit is realized to an excellent approximation for the electron (hole) band at $\nu = -1$ ($\nu = +1$), where the Hartree contribution from the filled bands gives rise to a single-particle term $\epsilon_0(\mathbf{k})$ which almost exactly cancels the interaction-generated Fock dispersion $\epsilon_F(\mathbf{k})$. Another motivation for starting with this limit is that it allows us to isolate the effects of band geometry and topology from those of the dispersion, which will be added later. In general, we will define an effective momentum space magnetic field $\mathcal{B} = \frac{2\pi C}{A_{\text{BZ}}}$ whose momentum space integral gives $2\pi C$. We define the corresponding magnetic length as $l_B = \sqrt{\frac{|\mathcal{B}|}{2\pi}}$. Unless otherwise stated, we will be using the parameters for TBG at the magic-angle $\theta = 1.0595°$ with chiral ratio $\kappa = 0.55$ and unscreened Coulomb interaction $V_{\mathbf{q}} = \frac{e^2}{2\epsilon\epsilon_0|\mathbf{q}|}$ with $\epsilon = 10$.

**From LLL to TBG Chern bands.** Let us start with the simplest possible Chern band, the lowest Landau level. To compare this model to Chern bands defined in momentum space, we define a real space unit cell that contains a single flux quantum so that $A_U = \frac{2\pi}{B}$ where $B$ is the real space magnetic field which is equal to the inverse of the momentum space magnetic field $\mathcal{B}$ since $B = \frac{2\pi}{A_U} = \frac{A_{\text{BZ}}}{2\pi} = \mathcal{B}^{-1}$. We can connect this LLL limit to chiral TBG using the results of ref. [37] which showed that the wavefunctions of the latter are equivalent to the LLL of a Dirac particle in the inhomogenous magnetic field $B_{\text{eff}}(\mathbf{r}) = B + B(\mathbf{r})$, for some specific $B(\mathbf{r})$ which averages to zero over the unit cell. We now consider a Dirac particle in magnetic field $B_{\text{eff}}(\mathbf{r}) = B + \eta B(\mathbf{r})$ where $\eta$ goes from 0 to 1, interpolating between the LLL and chiral TBG. We define $\Delta E$ to be the energy of the lowest $S = \frac{N-3}{2}$ eigenvalue of $H^{2e1h}$ relative to the energy of the lowest $S = \frac{N-1}{2}$. Note that in the thermodynamic limit, there is a continuum of $S = \frac{N-3}{2}$ states lying directly above the single particle $S = \frac{N-1}{2}$ state which implies that $\Delta E \leq 0$. Numerically, we expect to get a positive value for $\Delta E$ which scales to 0 with increasing system size whenever a bound state is absent. Throughout this work, we will set $\Delta E$ to zero in these cases. $\Delta E$ is shown in Fig. 1a as a function of $\eta$. We can see clearly that its value is negative for all $\eta$ indicating the formation of a bound state of an electron and a spin flip with binding energy around −1.5 meV for our choice of parameters. Although changing $\eta$ introduces variations in the Berry curvature distribution, the energy of the bound state is essentially independent of $\eta$.

**Tuning the chiral ratio.** Next, we introduce deviations from the chiral limit by considering the non-zero value for the chiral ratio $\kappa$. Finite $\kappa$ is known to alter the geometric properties of the bands and cause the Berry curvature to be concentrated at $\Gamma$[28,37]. For $\kappa \gtrsim 0.7 - 0.8$, the Berry curvature is very close to a delta function at $\Gamma$, which can be gauged away[38] leading to the loss of the band's topological character. As we can see in Fig. 1b, the bound state persists for all values of $\kappa \lesssim 0.8$, but its binding energy starts to approach zero as we approach the limit of very concentrated Berry curvature, hinting at its topological origin.

**Sublattice potential.** We can see the effect of topology more manifestly by considering a tuning parameter which alters the band topology by inducing a phase transition to a trivial Chern band. This is done by adding layer-dependent sublattice potential $\delta_{\text{top/bottom}}$, which can be physically realized from aligned hBN[5,39]. As was shown in ref. [40], the sublattice-polarized bands have vanishing Chern number when $\delta_{\text{top}} + \delta_{\text{bottom}}$ is close to 0, and finite Chern number $\pm 1$ otherwise. In Fig. 1c, we show $\Delta E$ as a function of $\delta_{\text{bottom}}$ for fixed $\delta_{\text{top}} = 10$ meV. We see that $\Delta E$ remains roughly constant on the topological side until we approach the transition, where it rapidly increases till it vanishes on the non-topological side.

**Gate screening.** So far, we have been considering unscreened Coulomb interaction relevant for TBG samples where the distance to the gate is much larger than the Moiré length scale. We will now consider

the effect of gate screening by taking $V_{\mathbf{q}}$ to be double-gate screened Coulomb interaction $V_{\mathbf{q}}(d) = \frac{e^2}{2\epsilon\epsilon_0|\mathbf{q}|}\tanh qd$ with $d$ denoting the gate distance. Since changing $d$ also alters the overall energy scale, we will find it more useful to measure energy in terms of the scale $E_C(d) = \frac{1}{2A}\sum_{\mathbf{q}}V_{\mathbf{q}}(d)e^{-\frac{|B|}{2}\mathbf{q}^2}$ which reduces to half the particle-hole gap for the LLL. The polaron binding energy is plotted as a function of $d$ in Fig. 1d which shows how $\Delta E$ starts decreasing when the gate distance is around 10 nm (roughly the Moiré scale) until it vanishes in the limit $d \to 0$. This is consistent with what is known about skyrmion energies which approaches the single-particle energy as the screening length is reduced. In fact, it was shown in ref. 22 that for the LLL, the energy of skyrmions of any size is the same as that of single-particle excitations for a delta potential, i.e., $d \to 0$.

## Topological electron-magnon coupling

To understand the existence of a bound state of an electron and a spin flip, it is instructive to rewrite the Hamiltonian $H^{2e1h}$ by labeling the Hilbert space of $S_z = -\frac{N-3}{2}$ in terms of an electron and a magnon excitation. The latter corresponds to the eigenmodes of the Hamiltonian in the space of single-spin flip operators

$$a^\dagger_{n,\mathbf{q}} = \sum_{\mathbf{k}} c^\dagger_{\mathbf{k},\uparrow} c_{\mathbf{k}+\mathbf{q},\downarrow}\phi_{n,\mathbf{q}}(\mathbf{k}), \quad \mathcal{H}a^\dagger_{n,\mathbf{q}}|\downarrow\rangle = \xi_{n,\mathbf{q}}a^\dagger_{n,\mathbf{q}}|\downarrow\rangle \tag{4}$$

where $\mathbf{q}$ belongs to the first BZ. The operators $a^\dagger_{n,\mathbf{q}}$ provide a complete $N$-dimensional orthonormal basis for spin flip operators, which can be used to represent any spin flip operator $c^\dagger_{\mathbf{k},\uparrow}c_{\mathbf{k}+\mathbf{q},\downarrow}$. The lowest energy state $n = 0$ corresponds to the Goldstone mode of the broken SU(2) spin symmetry whose dispersion satisfies $\xi_{0,\mathbf{q}} \to 0$ as $\mathbf{q} \to 0$.

Using this basis, we can construct a non-orthogonal basis of electron-magnon states as $|\mathbf{k}_0; \mathbf{q}, n\rangle = c^\dagger_{\mathbf{k}_0+\mathbf{q},\uparrow}a^\dagger_{n,\mathbf{q}}|\downarrow\rangle$ whose properties are discussed in detail in the methods section. In this basis, the Hamiltonian has the form

$$\mathcal{H}|\mathbf{k}_0; \mathbf{q}, n\rangle = \left[\xi_{n,\mathbf{q}} + \epsilon(\mathbf{k}_0+\mathbf{q})\right]|\mathbf{k}_0; \mathbf{q}, n\rangle \\ + \frac{1}{A}\sum_{\mathbf{q}'}V_{\mathbf{q}'}\lambda^*_{\mathbf{q}'}(\mathbf{k}_0+\mathbf{q})C^{nm}_{\mathbf{q},\mathbf{q}'}|\mathbf{k}_0; \mathbf{q}+\mathbf{q}', m\rangle \tag{5}$$

where $C^{nm}_{\mathbf{q},\mathbf{q}'}$ are defined as

$$C^{nm}_{\mathbf{q},\mathbf{q}'} = \sum_{\mathbf{k}}\phi^*_{m,\mathbf{q}+\mathbf{q}'}(\mathbf{k})\left[\lambda_{\mathbf{q}'}(\mathbf{k})\phi_{n,\mathbf{q}}(\mathbf{k}+\mathbf{q}') - \phi_{n,\mathbf{q}}(\mathbf{k})\lambda_{\mathbf{q}'}(\mathbf{k}+\mathbf{q})\right] \tag{6}$$

The meaning of the different terms in the Hamiltonian above is transparent. The first two terms correspond to the magnon and the electron dispersion, respectively. The last term corresponds to the matrix elements of the interaction between electron-magnon states with magnon momenta $\mathbf{q}$ and $\mathbf{q}'$. Low-lying excitations has their largest wieght in the lowest magnon branch $n = 0$. If we focus on this branch $n = m = 0$ and take the limit of small $\mathbf{q}$ and $\mathbf{q}'$, we find

$$C^{00}_{\mathbf{q},\mathbf{q}'} \approx i\mathbf{q}\wedge\mathbf{q}'B, \quad B = \frac{2\pi C}{A_{BZ}} \tag{7}$$

To see where this expression comes from, it is instructive to first consider the case of the LLL. One simplification we can do here is to unfold the BZ by extending $\mathbf{q}$ beyond the first BZ and removing the index $n$. The coefficient $C_{\mathbf{q},\mathbf{q}'}$ is then precisely the coefficient of the commutator of the GMP algebra $[\rho_{\mathbf{q}}, a^\dagger_{\mathbf{q}'}] = C_{\mathbf{q},\mathbf{q}'}a^\dagger_{\mathbf{q}+\mathbf{q}'}$[41], which in this case is equal to $2i\sin(\frac{B}{2}\mathbf{q}\wedge\mathbf{q}')$. For a general Chern band, the GMP algebra holds to linear order in $\mathbf{q}$ and $\mathbf{q}'$[42] with the prefactor given by $iB\mathbf{q}\wedge\mathbf{q}'$ which is precisely what we get in Eq. (7). A more detailed derivation of this result is given in the methods section.

Let us write a general eigenstate of (5) as

$$|\Psi\rangle = \sum_{n,\mathbf{q}} r_{n,\mathbf{q}}|\mathbf{k}_0; \mathbf{q}, n\rangle \tag{8}$$

Focusing on the $n = 0$ component in the small $\mathbf{q}$ limit, we see from (7) that the magnitude of the last term in the Hamiltonian (5) is maximized when connecting states $|\mathbf{k}_0; \mathbf{q}, n\rangle$ and $|\mathbf{k}_0; \mathbf{q}', n\rangle$ with $\mathbf{q}$ and $\mathbf{q}'$ orthogonal, i.e., if they are related by a $\pi/2$ rotation. Furthermore, due to the factor of $i$ in (7), we can make this term negative by choosing $r_{0,\mathbf{q}}$ to change its phase by $\pi/2$ upon rotating $\mathbf{q}$ by $\pi/2$. Thus, we can minimize this term by choosing $r_{0,\mathbf{q}} \propto e^{i\arg(q_x+iq_y)}$. In addition, since this term vanishes when $\mathbf{q}$ or $\mathbf{q}'$ vanish, the magnitude of $r_{0,\mathbf{q}}$ should not vanish too quickly with $\mathbf{q}$. We can see this by expanding the first two terms in (5) at small momenta $\xi_{0,\mathbf{q}} \sim l_B^2\rho\mathbf{q}^2$ and $\epsilon(\mathbf{k}_0+\mathbf{q}) \sim \frac{l_B^2}{m_{\text{eff}}}\mathbf{q}^2$. Then, if we assume that $r_{0,\mathbf{q}}$ decays for momenta larger than some cutoff $\Lambda$, we find that the first two terms in the Hamiltonian give a positive energy contribution of order $l_B^2(\rho + m_{\text{eff}}^{-1})\Lambda^2$ whereas the last term gives a negative contribution of order $E_C l_B^3\Lambda^3$. Thus, a bound state has to have a finite extent in momentum of at least $\Lambda \sim \frac{\rho + m^{-1}}{E_C l_B}$. This is verified by plotting $r_{0,\mathbf{q}}$ for both the LLL and chiral TBG in Fig. 2. We see that $|r_{0,\mathbf{q}}|$ decays in $\mathbf{q}$ within the first BZ and that $\arg r_{0,\mathbf{q}}$ winds by $2\pi$ around the $\Gamma$ point. For the LLL case, due to continuous magnetic translation, we can unfold the BZ and write the variational state $r_{\mathbf{q}} = e^{-\frac{\xi}{2}|\mathbf{q}| + i\arg(q_x+iq_y)}$ whose overlap with the numerically obtained solution exceeds 99% for appropriately chosen $\xi$ (see supplemental material for details).

## Finite quasiparticle dispersion

Let us now consider the limit of dispersive quasiparticle bands. Motivated by the energetics of TBG bands, we will choose $\epsilon_0(\mathbf{k}) = v\epsilon_H(\mathbf{k})$ where $\epsilon_H(\mathbf{k}) = \frac{1}{A}\sum_{\mathbf{G}}V_{\mathbf{G}}\lambda_{\mathbf{G}}(\mathbf{k})\sum_{\mathbf{k}'}\lambda_{-\mathbf{G}}(\mathbf{k}')$ is the Hartree potential[11,28,30,43–45]. This form of the dispersion directly allows us to compare our results to TBG since the quasiparticle dispersion $\epsilon_\nu(\mathbf{k}) = v\epsilon_H(\mathbf{k}) + \epsilon_F(\mathbf{k})$ describes the dispersion of electron (hole) bands for a correlated insulator at integer filling $\nu$ ($-\nu$) in the independent Chern sector approximation[12,44]. Although the expression for $\epsilon_\nu(\mathbf{k})$ is only valid for integer $\nu$, we will choose to take $\nu$ to be a continuous variable which allows us to continuously tune the band dispersion. It also makes our results less sensitive to uncertainty in model parameters. For instance, while our model is particle-hole symmetric, we can phenomenologically incorporate particle-hole asymmetry by shifting $\nu$, which approximately captures the effects discussed in ref. 46. $\epsilon_H(\mathbf{k})$ is characterized by a dip at the $\Gamma$ point and thus have a qualitatively similar shape to the Fock term $\epsilon_F(\mathbf{k})$ up to overall scaling. Thus for $\nu > 0$, the two terms add, leading to a large bandwidth, while for $\nu < 0$, they subtract, leading to a reduced bandwidth[11,30,44,45]. The minimum bandwidth is realized for $\nu \approx -1$ to $-1.5$ depending on the value of $\kappa$ as shown by the dashed line in Fig. 3a. The details of the dispersion are reviewed in the methods section.

We can investigate the existence of a bound state as a function of dispersion parameterized by $\nu$. We find that there is a critical value $\nu_c$, indicated by the solid line in Fig. 3a, such that a bound state exists iff $\nu < \nu_c$. We see that this value is always negative and ranges from around $-0.5$ in the chiral limit to around $-1.2$ at $\kappa = 0.7$. The implication for TBG is that, generally, polaron formation is favored on electron (hole) doping the $\nu < 0$ ($\nu > 0$) insulators, i.e., doping towards neutrality. On doping away from neutrality, we always find the single-particle excitations to be the lower in energy. We note that for $\nu$ sufficiently large and negative, the Hartree term dominates and we get a peak rather than a dip at $\Gamma$. In this case, we always find a polaron-bound state even though the bandwidth can be quite large. This rather surprising results can be explained by examining the wavefunction of the $\Gamma$ polaron in

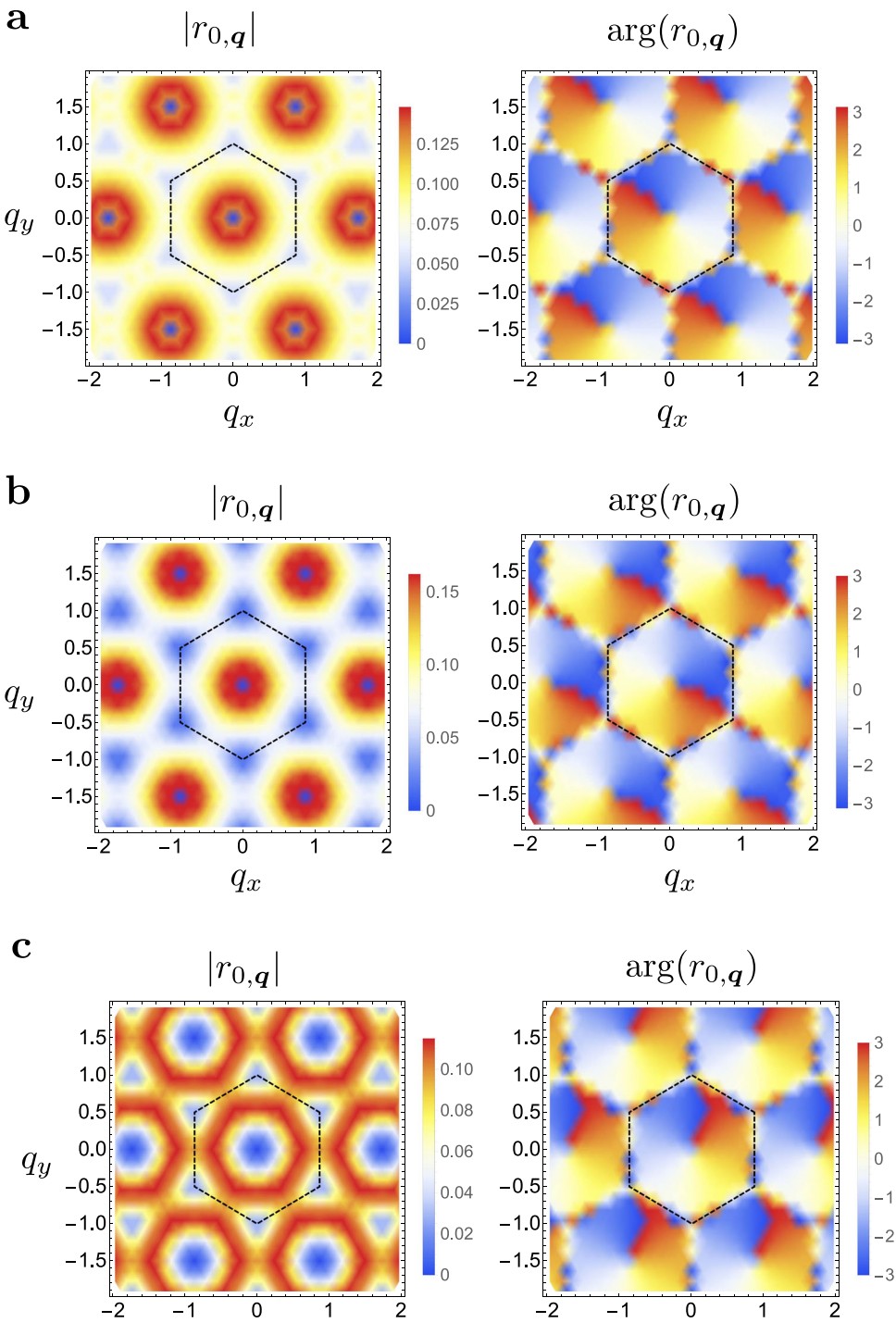

**Fig. 2 | Polaron wavefunctions.** Color plot of the polaron wavefunction as a function of the magnon momentum **q** (cf. Eq. (8)) for **a** the LLL, TBG bands (**b**) with flat quasiparticle dispersion, and **c** for the electron (hole) doping the $\nu = -3$ ($\nu = +3$) insulator.

Fig. 2c. We see that, compared to the case of flat quasiparticle dispersion, the wavefunction has suppressed weight at small **q** enabling it to avoid the energetically costly region around Γ.

This suggests that the formation of a bound state is mostly sensitive to the effective mass at the bottom of the band, which is relatively large at the band minimum for $\nu$ large and negative, rather than the total bandwidth. The effective mass has the added advantage of being an experimentally accessible quantity, e.g., from quantum oscillations[1,2], allowing us to make phenomenological comparisons with experiments that are not tied to theory parameters. In Fig. 3b, we

show the effective mass for different values $(\nu, \kappa)$ and compare it to the phase diagram in Fig. 3a. We find that, quite remarkably, the phase boundary where the polaron-bound state is lost can be described very well by the expression $m_e/m_{\text{eff}} \approx 3$ as shown in Fig. 3b. This value is not far off the experimentally extracted value $m_{\text{eff}}/m_e \approx 0.2$–0.3 from quantum oscillations for small hole doping of the $\nu = -2$ state where superconductivity was first seen[1]. This suggests that the experimental regime for superconductivity can be close to the phase boundary where a bound state would form even when the lowest charged excitation are single-particle-like.

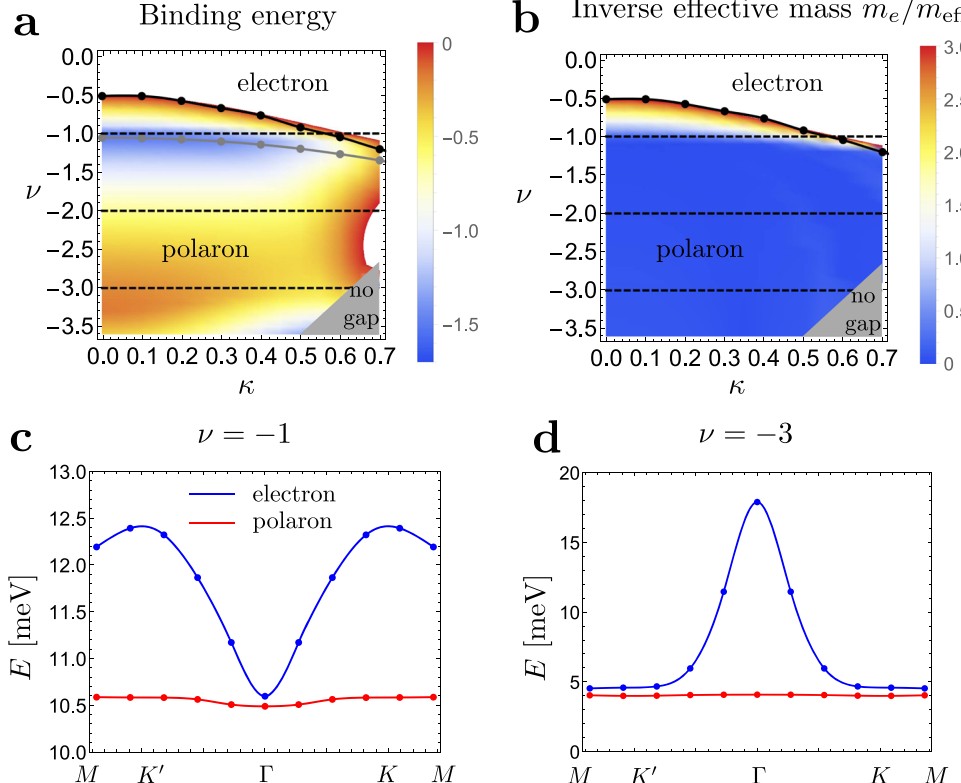

**Fig. 3 | Polaron energetics in twisted bilayer graphene. a** Binding energy as a function of "filling" $\nu$ and chiral ratio $\kappa$. The dashed lines indicate integer fillings $\nu$ where our dispersion matches the HF dispersion for electron doping. A bound state is only found for $\nu < 0$, indicating doping towards neutrality. The solid black line indicates the boundary where the bound state is lost, the dashed gray line corresponds to the minimum bandwidth, and the gray shaded area is where bands overlap and our analysis becomes invalid. **b** The corresponding inverse effective mass at the bottom of the band. The phase boundary is well approximated by the value $m_e/m_{eff} \approx 3$. **c, d** are the spectra for electron doping the $\nu = -1$ and $\nu = -3$ insulators, respectively.

Finally, we investigate the dispersion of the spin polaron state in the limit when it is the lowest energy excitation. We consider two cases: (i) electron (hole) doping the $\nu = -1$ ($\nu = 1$) insulator where the dispersion minimum is at $\Gamma$ and (ii) electron (hole) doping the $\nu = -3$ ($\nu = +3$) insulator where the dispersion *maximum* is at $\Gamma$. The resulting dispersion is shown in Fig. 3c, d. We see that the polarons have much flatter bands with significantly larger effective mass that can be as large as 30 times the electron's effective mass.

## Discussion
### Implications for TBG
Before discussing potential implications for our results, let us point out a few caveats. First, the lowest energy charged excitations we obtain here are just among single particle states and those dressed by a single-spin flip. This does not rule out the possibility that states with more spin flips are lower energy excitations (which is known to be the case for the LLL[14,15]) even in the cases the spin polaron is not lower in energy than single particle excitations. This means that our results establish a parameter regime where the electron is not the lowest energy charge $e$ excitation but cannot make a strong statement about the precise number of spin flips or the nature of charge $e$ excitations outside that regime. We note, however, that once we are in the domain of stability of polarons, we expect significant modification of the physics compared to single electrons, even if we cannot determine precisely the number of of bound spin flips. In other words, we believe that once spin polarons are formed, no matter the precise size, their properties are well approximated by the simplest nontrivial one studied here e.g., they will have very flat dispersion. We note here that by applying a Zeeman field, the

energy of a spin polaron with $n$ spin flips is increased by $\Delta E_{S_z = (N-1)/2-n} = (n+1/2)E_{Zeeman}$. This disfavors larger polarons/skyrmions and it can be chosen such that the single-spin flip polaron $n = 1$ is the lowest energy excitation if it is already lower in energy than the electron at zero fields provided that the energy as a function of the number of spin $n$ flips is a convex function. This means that the energy difference between the single-spin flip polaron and the electron exceeds the difference between the polaron with $n + 1$ spin flips and that with $n$ spin flips for any $n > 0$. $E_{Zeeman}$ can be realized via in-plane magnetic field if these are spin skyrmions or via sublattice potential if these are pseudospin skyrmions.

Second, our analysis focused on a single Chern sector. While we took into account Hartree–Fock interaction-generated dispersion, which is mainly diagonal in the Chern index, we have neglected the part of the interaction and the dispersion connecting different Chern sectors. The inter-Chern part of the interaction vanishes in the chiral limit and is otherwise a relatively small correction that gives rise to an inter-Chern pseudospin coupling $\lambda \approx 0.4 - 0.6$, which is antiferromagnetic in-plane and ferromagnetic out-of-plane[12,20,38]. This gives rise to a Zeeman term with $E_{Zeeman} = \lambda$ which only affects pseudospin polarons. The inter-Chern part of the dispersion comes from the intrinsic dispersion of the BM model as well as the subtraction scheme employed when projecting out the remote bands to avoid double counting (see refs. 11, 12, 28, 47). It takes the form of tunneling between opposite Chern sectors, which influences physics in several ways. First, the electron and polaron are no longer distinguished by their spin quantum number since the spin is no longer conserved in a given Chern sector, and thus, they can tunnel into each other. Second, such tunneling perturbatively generates an antiferromagnetic

"superexchange" coupling $J \approx 0.5 - 1$ meV[12,20,38] between the Chern sectors, which alters the energetics of magnons and plays a crucial role in the skyrmion pairing scenario[20,21]. We expect these terms to favor charge $2e$ polaron *pairs* (bipolarons) over single charge $e$ polarons since they act as a Zeeman field $E_{Zeeman} = J$ for spin or pseudospin polarons but do not affect the energy of polaron pairs. We leave a detailed analysis of the effect of these terms in future works. Finally, we have only focused on SU(2) polarons which assume there are only two active bands while the remaining bands are frozen. This approach can be phenomenologically justified by the observation of flavor polarization 'cascade' features at relatively high temperatures[48,49] suggesting these polarized flavor degrees of freedom becomes frozen at low temperature. However, based on energetics alone, we cannot role out more complicated SU(4) skyrmion/polaron textures.

Our findings suggest the following picture for charge excitations in TBG: (i) On doping a correlated insulator at integer $\nu \neq 0$ towards charge neutrality, charge enters the system as polarons or large skyrmions. This is consistent with the observed absence of quantum oscillations for this doping range[1,2,50] and also explains the rapid loss of flavor (spin/valley) polarization (cascade transition)[48,49] with doping. Combined with the observations that polarons are disfavored by reducing the screening length, this leads to the prediction that the cascade features should become weaker as the screening length to the gate is reduced[4,51]. (ii) On doping away from neutrality, charge likely enters the system as single-particle excitations consistent with the observation of Landau fans. Finally, Although we have focused on charge $e$ excitations, let us make a few observations about pairing, i.e. the charge $2e$ excitations. Pairing between stable spin-polarons, the analog of the skyrmion pairing mechanism proposed by the authors in ref. 20 (see also refs. 21, 52) can be naturally associated with doping towards neutrality where superconductivity has been observed in some samples[1,3,4]. On the other hand, on doping away from neutrality (where superconductivity is seemingly more ubiquitous), spin polarons can remain relevant as finite energy long-lived excitations whose pairing correlations can be induced to the electrons, even when they are not the lowest charge excitations. This suggests a BEC-to-BCS scenario with increasing dispersion where the bound state is lost while superconductivity persists[53]. A detailed theory of spin-polaron pairing will be the topic of future work.

In summary, we have identified a general tendency for the formation of a polaron-bound state between an electron and a spin flip in a Chern band that is purely topological in origin. We have studied the formation of such bound states over a wide range of parameters for the Chern bands of twisted bilayer graphene. This lead us to identify the experimental parameter range where spin polarons are formed and discuss their possible experimental consequences. Our results highlight the surprising fact that although the ground state is well approximated by a Slater determinant, a description in terms of electron-like single-particle excitations, whether approximate or exact, is insufficient to describe the charge physics in Chern bands. Furthermore, our analysis serves as a bridge between real space skyrmion textures and single-particle excitations.

**Note.** We would like to point out a parallel work[54] which gives an extensive report of the energetics of TBG skyrmions, both charge "$e$" and charge "$2e$", using variational Hartree−Fock. The results of that work, which is suited to study the limit of large skyrmions, is complementary to our momentum space approach. In addition, After the appearance of our work on arXiv, ref. 55 appeared, which also studied spin polarons in the specific context of TBG and focused on the limit of short screening gate distance. In the chiral limit, their calculations agree with ours for the same parameters (after accounting for a difference in the notation for screening length, which is defined in ref. 55 as twice the distance between gate and sample). Away from the chiral limit, our work, and ref. 55 adopt different approximations.

## Methods

### Explicit form of the Hamiltonian and total spin operators

The explicit form of the Hamiltonian $\mathcal{H}^{2e1h}$ can be easily obtained from the action of the Hamiltonian $\mathcal{H}$, Eq. (1), on $|\mathbf{k}_{e1}, \mathbf{k}_{e2}, \mathbf{k}_h\rangle$, defined in Eq. (3), using the commutation relations

$$\left[\delta\rho_{\mathbf{q}}, c^\dagger_{\sigma,\mathbf{k}}\right] = \lambda^*_{-\mathbf{q}}(\mathbf{k})c^\dagger_{\sigma,\mathbf{k}-\mathbf{q}}, \quad \left[\delta\rho_{\mathbf{q}}, c_{\sigma,\mathbf{k}}\right] = -\lambda_{\mathbf{q}}(\mathbf{k})c_{\sigma,\mathbf{k}+\mathbf{q}} \quad (9)$$

leading to

$$\begin{aligned}
\mathcal{H}|\mathbf{k}_{e1}, \mathbf{k}_{e2}, \mathbf{k}_h\rangle = &\left[\epsilon_e(\mathbf{k}_{e1}) + \epsilon_e(\mathbf{k}_{e2}) + \epsilon_h(\mathbf{k}_h)\right]|\mathbf{k}_{e1}, \mathbf{k}_{e2}, \mathbf{k}_h\rangle \\
&+ \frac{1}{A}\sum_{\mathbf{q}} V_{\mathbf{q}}\Big\{\lambda^*_{\mathbf{q}}(\mathbf{k}_{e1})\lambda^*_{-\mathbf{q}}(\mathbf{k}_{e2})|\mathbf{k}_{e1}+\mathbf{q}, \mathbf{k}_{e2}-\mathbf{q}, \mathbf{k}_h\rangle \\
&- \lambda^*_{\mathbf{q}}(\mathbf{k}_{e1})\lambda_{\mathbf{q}}(\mathbf{k}_h)|\mathbf{k}_{e1}+\mathbf{q}, \mathbf{k}_{e2}, \mathbf{k}_h+\mathbf{q}\rangle \\
&- \lambda^*_{\mathbf{q}}(\mathbf{k}_{e2})\lambda_{\mathbf{q}}(\mathbf{k}_h)|\mathbf{k}_{e1}, \mathbf{k}_{e2}+\mathbf{q}, \mathbf{k}_h+\mathbf{q}\rangle\Big\}
\end{aligned}$$
$$(10)$$

To express the total spin operator in the basis $|\mathbf{k}_{e1}, \mathbf{k}_{e2}, \mathbf{k}_h\rangle$, we start the standard expression

$$S^2 = S_x^2 + S_y^2 + S_z^2, \quad (11)$$

$$S_x = \frac{1}{2}\sum_{\mathbf{k}}\left(c^\dagger_{\mathbf{k},\uparrow}c_{\mathbf{k},\downarrow} + c^\dagger_{\mathbf{k},\downarrow}c_{\mathbf{k},\uparrow}\right), \quad (12)$$

$$S_y = -\frac{i}{2}\sum_{\mathbf{k}}\left(c^\dagger_{\mathbf{k},\uparrow}c_{\mathbf{k},\downarrow} - c^\dagger_{\mathbf{k},\downarrow}c_{\mathbf{k},\uparrow}\right), \quad (13)$$

$$S_z = \frac{1}{2}\sum_{\mathbf{k}}\left(c^\dagger_{\mathbf{k},\uparrow}c_{\mathbf{k},\uparrow} - c^\dagger_{\mathbf{k},\downarrow}c_{\mathbf{k},\downarrow}\right) \quad (14)$$

The state $|\mathbf{k}_{e1}, \mathbf{k}_{e2}, \mathbf{k}_h\rangle = c^\dagger_{\mathbf{k}_{e1},\uparrow}c^\dagger_{\mathbf{k}_{e2},\uparrow}c_{\mathbf{k}_h,\downarrow}|\downarrow\rangle$ is an $S_z$ eigenstate with eigenvalue $-\frac{N-3}{2}$. The action of $S_x$ and $S_y$ can be obtained using the commutations relations

$$\left[S_x, c^\dagger_{\mathbf{k},\uparrow}\right] = \frac{1}{2}c^\dagger_{\mathbf{k},\downarrow}, \qquad \left[c^\dagger_{\mathbf{k},\downarrow}, S_x\right] = -\frac{1}{2}c^\dagger_{\mathbf{k},\uparrow}, \quad (15)$$

$$\left[S_x, c_{\mathbf{k},\downarrow}\right] = -\frac{1}{2}c_{\mathbf{k},\uparrow}, \qquad \left[c_{\mathbf{k},\uparrow}, S_x\right] = \frac{1}{2}c_{\mathbf{k},\downarrow} \quad (16)$$

$$\left[S_y, c^\dagger_{\mathbf{k},\uparrow}\right] = \frac{i}{2}c^\dagger_{\mathbf{k},\downarrow}, \qquad \left[c^\dagger_{\mathbf{k},\downarrow}, S_y\right] = \frac{i}{2}c^\dagger_{\mathbf{k},\uparrow}, \quad (17)$$

$$\left[S_y, c_{\mathbf{k},\downarrow}\right] = -\frac{i}{2}c_{\mathbf{k},\uparrow}, \qquad \left[c_{\mathbf{k},\uparrow}, S_y\right] = -\frac{i}{2}c_{\mathbf{k},\downarrow} \quad (18)$$

which leads after straightforward but tedious calculations to

$$\begin{aligned}
S_x^2|\mathbf{k}_{e1}, \mathbf{k}_{e2}, \mathbf{k}_h\rangle = \frac{1}{4}\Big[&-3|\mathbf{k}_{e1}, \mathbf{k}_{e2}, \mathbf{k}_h\rangle \\
&+ 2\sum_{\mathbf{k}}\left(-\delta_{\mathbf{k}_{e1},\mathbf{k}_h}|\mathbf{k}_{e2}, \mathbf{k}, \mathbf{k}\rangle + \delta_{\mathbf{k}_{e2},\mathbf{k}_h}|\mathbf{k}_{e1}, \mathbf{k}, \mathbf{k}\rangle\right)\Big] \quad (19) \\
&+ c^\dagger_{\mathbf{k}_{e1},\uparrow}c^\dagger_{\mathbf{k}_{e2},\uparrow}c_{\mathbf{k}_h,\uparrow}S_x^2|\downarrow\rangle
\end{aligned}$$

$$S_y^2|\mathbf{k}_{e1},\mathbf{k}_{e2},\mathbf{k}_h\rangle = -\frac{1}{4}\Big[3|\mathbf{k}_{e1},\mathbf{k}_{e2},\mathbf{k}_h\rangle$$
$$+ 2\sum_{\mathbf{k}}\Big(\delta_{\mathbf{k}_{e1},\mathbf{k}_h}|\mathbf{k}_{e2},\mathbf{k},\mathbf{k}\rangle - \delta_{\mathbf{k}_{e2},\mathbf{k}_h}|\mathbf{k}_{e1},\mathbf{k},\mathbf{k}\rangle\Big)\Big] \qquad (20)$$
$$+ c_{\mathbf{k}_{e1},\uparrow}^\dagger c_{\mathbf{k}_{e2},\uparrow}^\dagger c_{\mathbf{k}_h,\downarrow} S_y^2|\downarrow\rangle$$

Which leads to

$$S^2|\mathbf{k}_{e1},\mathbf{k}_{e2},\mathbf{k}_h\rangle = \left(\frac{N-1}{2}\right)\left(\frac{N-3}{2}\right)|\mathbf{k}_{e1},\mathbf{k}_{e2},\mathbf{k}_h\rangle$$
$$+ \hat{M}|\mathbf{k}_{e1},\mathbf{k}_{e2},\mathbf{k}_h\rangle, \qquad (21)$$

$$\hat{M}|\mathbf{k}_{e1},\mathbf{k}_{e2},\mathbf{k}_h\rangle = \sum_{\mathbf{k}}\Big(-\delta_{\mathbf{k}_{e1},\mathbf{k}_h}|\mathbf{k}_{e2},\mathbf{k},\mathbf{k}\rangle + \delta_{\mathbf{k}_{e2},\mathbf{k}_h}|\mathbf{k}_{e1},\mathbf{k},\mathbf{k}\rangle\Big) \qquad (22)$$

It is easy to verify that the operator $\hat{M}$ defined on the second line satisfies $\hat{M}^2 = (N-1)\hat{M}$, hence its eigenvalues are 0 and $N-1$. The former yields $S^2$ eigenvalue $\left(\frac{N-1}{2}\right)\left(\frac{N-3}{2}\right)$ which corresponds to a total spin $S = \frac{N-3}{2}$ whereas the latter yields $S^2$ eigenvalue $\left(\frac{N-1}{2}\right)\left(\frac{N+1}{2}\right)$ which corresponds to a total spin $S = \frac{N-1}{2}$.

## Properties of the electron-magnon basis
In this section, we discuss the properties of the electron-magnon basis introduced in the main text, repeated here for completeness

$$|\mathbf{k}_0;\mathbf{q},n\rangle = c_{\mathbf{k}_0+\mathbf{q},\uparrow}^\dagger a_{n,\mathbf{q}}^\dagger|\downarrow\rangle \qquad (23)$$

First, note that the state $|\mathbf{k}_0;0,0\rangle$ is a single-particle state with $S = \frac{N-1}{2}$ since $a_{0,\mathbf{q}=0}^\dagger$ is simply the generator of a uniform spin rotation which increases $S_z$ of the ground state by 1. We note that the basis (23) contains $N^2$ states for a given $\mathbf{k}_0$ which is more than the size of the Hilbert space given by $\frac{N(N-1)}{2}$. The reason is this basis is that it is not orthonormal. Instead, the overlap of states is given by

$$g_{\mathbf{q},\mathbf{q}'}^{nm}(\mathbf{k}_0) = \langle\mathbf{k}_0;\mathbf{q},n|\mathbf{k}_0;\mathbf{q}',m\rangle$$
$$= \delta_{mn}\delta_{\mathbf{q},\mathbf{q}'} - \phi_{m\mathbf{q}'}^*(\mathbf{k}_0+\mathbf{q})\phi_{n\mathbf{q}}(\mathbf{k}_0+\mathbf{q}') \qquad (24)$$

This overlap can be identified with the matrix elements of the operator $1-\hat{F}$ where $\hat{F}$ is the operator that exchanges two electrons. The $N^2$ basis states for a given $\mathbf{k}_0$ include $\frac{N(N-1)}{2}$ fermionic (antisymmetric) states with $g(\mathbf{k}_0)$ eigenvalues 2 and $\frac{N(N+1)}{2}$ bosonic (symmetric) states with $g(\mathbf{k}_0)$ eigenvalues 0. Since the exchange operator $\hat{F}$ commutes with the Hamiltonian, we can obtain the physical Hilbert space (3) simply by restricting to the eigenstates of the Hamiltonian with $g(\mathbf{k}_0)$ eigenvalue 2.

## Derivation of the topological electron-magnon coupling at small momenta
Our purpose in this section is to derive the form of the electron-magnon coupling at small momenta, Eq. (7). The magnon creation operator is defined in Eq. (4), repeated here for completeness

$$a_{n,\mathbf{q}}^\dagger = \sum_{\mathbf{k}} c_{\uparrow,\mathbf{k}}^\dagger c_{\downarrow,\mathbf{k}+\mathbf{q}}\phi_{n,\mathbf{q}}(\mathbf{k}) \qquad (25)$$

where $\phi_{n,\mathbf{q}}(\mathbf{k})$ is the complete orthonormal set of eigenfunctions of the soft mode Hamiltonian defined as

$$\mathcal{H}_{\mathbf{q}}(\mathbf{k}',\mathbf{k}) = \left\langle c_{\downarrow,\mathbf{k}'+\mathbf{q}}^\dagger c_{\uparrow,\mathbf{k}'}\mathcal{H}_V c_{\uparrow,\mathbf{k}}^\dagger c_{\downarrow,\mathbf{k}+\mathbf{q}}\right\rangle, \qquad (26)$$

$$\sum_{\mathbf{k}'}\mathcal{H}_{\mathbf{q}}(\mathbf{k},\mathbf{k}')\phi_{n,\mathbf{q}}(\mathbf{k}') = \xi_{n,\mathbf{q}}\phi_{n,\mathbf{q}}(\mathbf{k}) \qquad (27)$$

We notice that gauge invariance requires that $\phi_{n,\mathbf{q}}(\mathbf{k})$ transforms the same way as $\lambda_{\mathbf{q}}(\mathbf{k})$ under gauge transformations. That is, under $c_{\mathbf{k},\sigma}\mapsto c_{\mathbf{k},\sigma}e^{i\theta_{\mathbf{k}}}$, $\phi_{n,\mathbf{q}}(\mathbf{k})\mapsto e^{-i[\theta_{\mathbf{k}+\mathbf{q}}-\theta_{\mathbf{k}}]}\phi_{n,\mathbf{q}}(\mathbf{k})$. This means we can define a gauge invariant $\tilde{\phi}_{n,\mathbf{q}}(\mathbf{k})$ via

$$\phi_{n,\mathbf{q}}(\mathbf{k}) = \tilde{\lambda}_{\mathbf{q}}(\mathbf{k})\tilde{\phi}_{n,\mathbf{q}}(\mathbf{k}), \qquad \tilde{\lambda}_{\mathbf{q}}(\mathbf{k}) = \frac{\lambda_{\mathbf{q}}(\mathbf{k})}{|\lambda_{\mathbf{q}}(\mathbf{k})|} \qquad (28)$$

where we used the phase of the form factor $\tilde{\lambda}_{\mathbf{q}}(\mathbf{k})$ rather than its full value to maintain the normalization of the wavefunctions. It is easy to show that in the limit $\mathbf{q}\to 0$, $\phi_{0,\mathbf{q}}(\mathbf{k})\to\frac{1}{\sqrt{N}}$. This is nothing but the statement that the Goldstone mode in the limit of long wavelength reduces to the spin raising operator. Thus, we can write

$$\tilde{\phi}_{0,\mathbf{q}}(\mathbf{k}) \approx \frac{1}{\sqrt{N}}[1 + i\mathbf{q}\cdot\mathbf{v}(\mathbf{k}) + O(\mathbf{q}^2)] \qquad (29)$$

Crucially, we can show that the Hamiltonian $\tilde{\mathcal{H}}_{\mathbf{q}}(\mathbf{k}',\mathbf{k}) = \tilde{\lambda}_{\mathbf{q}}^*(\mathbf{k}')\mathcal{H}_{\mathbf{q}}(\mathbf{k}',\mathbf{k})\tilde{\lambda}_{\mathbf{q}}(\mathbf{k})$ is periodic and smooth in $\mathbf{k}$, and so is $\tilde{\phi}_{n,\mathbf{q}}(\mathbf{k})$ and $\mathbf{v}(\mathbf{k})$. This can be seen by writing the transformed Hamiltonian $\tilde{H}_{\mathbf{q}}(\mathbf{k},\mathbf{k}') = \tilde{\lambda}_{\mathbf{q}}^*(\mathbf{k})H_{\mathbf{q}}(\mathbf{k},\mathbf{k}')\tilde{\lambda}_{\mathbf{q}}(\mathbf{k}')$ and noting that it only depends on gauge-invariant combinations of $\tilde{\lambda}_{\mathbf{q}}(\mathbf{k})$ which can be written in terms of the Berry curvature, which is periodic and smooth in $\mathbf{k}$. Substituting (28) and (29) in Eq. 6 in the main text and using the small $\mathbf{q}$ expansion of the form factor $\lambda_{\mathbf{q}}(\mathbf{k}) \approx 1 + i\mathbf{q}\cdot\mathbf{A}(\mathbf{k}) + O(\mathbf{q}^2)$ yields

$$C_{\mathbf{q},\mathbf{q}'}^{00} = \frac{i}{N}\sum_{\mathbf{k}}\Big\{q'_\mu[A^\mu(\mathbf{k}) - A^\mu(\mathbf{k}+\mathbf{q})]$$
$$+ q_\mu[A^\mu(\mathbf{k}+\mathbf{q}') - A^\mu(\mathbf{k})] + q_\mu[v^\mu(\mathbf{k})$$
$$= iq_\mu q'_\nu\int\frac{d^2\mathbf{k}}{A_{BZ}}[\partial_\mu A_\nu(\mathbf{k}) - \partial_\nu A_\mu(\mathbf{k})]$$
$$= i\mathbf{q}\wedge\mathbf{q}'\int\frac{d^2\mathbf{k}}{A_{BZ}}\Omega(\mathbf{k}) = i\frac{2\pi C}{A_{BZ}}\mathbf{q}\wedge\mathbf{q}' \qquad (30)$$

On going from the first to the second line, we used the periodicity of $\mathbf{v}(\mathbf{k})$ to shift the momentum summation leading to $\sum_{\mathbf{k}}v^\mu(\mathbf{k}+\mathbf{q}') - \sum_{\mathbf{k}}v^\mu(\mathbf{k}) = 0$ (notice that this does not work for $A_\mu(\mathbf{k})$, which cannot be periodic in a band with finite Chern number). In the last equality, we used the definition of the Chern number $\int d^2\mathbf{k}\,\Omega(\mathbf{k}) = 2\pi C$.

## Dispersion
Here, we discuss some details about the dispersion we used in the main text. At integer fillings and if we ignore the inter-Chern part of the dispersion (the decoupled Chern sector approximation), the single particle dispersion is given[29,30] by diagonalizing the Hartree–Fock Hamiltonian[12,28] which takes the form

$$H_{HF}[Q](\mathbf{k}) = \frac{1}{2A}\sum_{\mathbf{G}}V_{\mathbf{G}}\Lambda_{\mathbf{G}}(\mathbf{k})\sum_{\mathbf{k}'}\mathrm{tr}\Lambda_{-\mathbf{G}}(\mathbf{k}')Q_{\mathbf{k}}$$
$$- \frac{1}{2A}\sum_{\mathbf{q}}V_{\mathbf{q}}\Lambda_{\mathbf{q}}(\mathbf{k})Q_{\mathbf{k}+\mathbf{q}}\Lambda_{\mathbf{q}}(\mathbf{k})^\dagger \qquad (31)$$

Here, $Q_{\mathbf{k}}$ is a matrix with eigenvalues $\pm 1$ describing a Slater determinant state such that $\pm 1$ correspond to full/empty electronic states. $\Lambda_{\mathbf{q}}(\mathbf{k})$ is a matrix for form factors with spin ($s$), sublattice ($\sigma$), and valley ($\tau$) indices which can be transformed into a Chern ($\gamma$), spin ($s$), and pseudospin ($\eta$) basis (see refs. 12, 20, 38). Since our analysis focuses on a single Chern sector, we are going to neglect the Chern off-diagonal terms in the form factor $\Lambda$, which was shown in ref. 12 to be relatively small (they vanish identically in the chiral limit). In this limit, $\Lambda_{\mathbf{q}}(\mathbf{k})$ takes the simple form $\Lambda_{\mathbf{q}}(\mathbf{k}) = s_0\otimes\eta_0\otimes\mathrm{diag}(\lambda_{\mathbf{q}}(\mathbf{k}),\lambda_{\mathbf{q}}^*(\mathbf{k}))_\gamma$ where $\lambda_{\mathbf{q}}(\mathbf{k})$ are the form factors for a single Chern band. At integer filling $\nu$ and

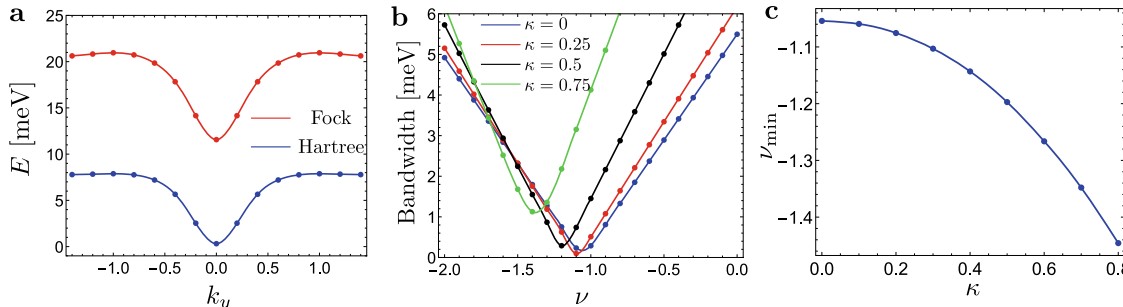

**Fig. 4 | Details of the approximate Hartree–Fock dispersion. a** Hartree $\epsilon_H(\mathbf{k})$ and Fock $\epsilon_F(\mathbf{k})$ dispersion. **b** Bandwidth for the dispersion $\epsilon_\nu(\mathbf{k})$ as a function of $\nu$ for different values of $\kappa$. **c** Value of $\nu$ for which the bandwidth is minimum as a function of $\kappa$.

ignoring inter-Chern dispersion, the family of the ground state is described $Q_\mathbf{k}$ can be chosen to be $\mathbf{k}$-independent and to satisfy $\mathrm{tr}Q = 2\nu$ and $[Q, \Lambda_\mathbf{q}(\mathbf{k})] = 0$, which is equivalent to the condition that $Q$ is Chern-diagonal, i.e., $[Q, \gamma_z] = 0$[12]. Under these conditions, the Hamiltonian simplifies to

$$H_{\mathrm{HF}}[Q](\mathbf{k}) = \epsilon_{\nu,\pm}(\mathbf{k}) = \pm\epsilon_F(\mathbf{k}) + \nu\epsilon_H(\mathbf{k}), \tag{32}$$

$$\epsilon_H(\mathbf{k}) = \frac{1}{A}\sum_\mathbf{G} V_\mathbf{G}\lambda_\mathbf{G}(\mathbf{k})\sum_{\mathbf{k}'}\lambda_{-\mathbf{G}}(\mathbf{k}'), \tag{33}$$

$$\epsilon_F(\mathbf{k}) = \frac{1}{2A}\sum_\mathbf{q} V_\mathbf{q}|\lambda_\mathbf{q}(\mathbf{k})|^2 \tag{34}$$

where the positive (negative) sign is for the electron (hole) bands. We note that $\epsilon_{-\nu,\pm} = -\epsilon_{\nu,\mp}$ due to particle-hole symmetry so that electron (hole) bands on the $\nu > 0$ side map to hole (electron) bands on the $\nu < 0$ side. That is, doping away from charge neutrality is the same whether for positive and negative $\nu$ and similarly for doping towards neutrality. The Hartree and Fock potentials are plotted in Fig. 4a, and we can see that both are characterized by a dip at $\Gamma$. Thus, for doping away from neutrality, the two are going to add, while on doping towards neutrality, they subtract. In the main text, we used $\nu$ as an interpolation parameter that also takes non-integer values as a proxy for tuning the bandwidth. We can see in Fig. 4b the bandwidth as a function of $\nu$, and we see that there is a minimum in the range $\nu \in [-1.5, -1]$ depending on the chiral ratio $\kappa$. The value of $\nu = \nu_{\min}$ for which the bandwidth is minimum is shown in Fig. 4c. We note that for $\nu > \nu_{\min}$, the band minimum is at $\Gamma$ whereas for $\nu < \nu_{\min}$, the band maximum is at $\Gamma$.

## Data availability
The data that support the findings of this study are available from the corresponding author upon request.

## Code availability
All numerical codes in this paper are available upon request to the authors.

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

## Acknowledgements

We acknowledge stimulating discussions with Mike Zaletel, Senthil Todadri, Herb Fertig, Patrick Ledwith, and Dan Parker. We are grateful to Yves Kwan, Nick Bultinck, and Sid Parameswaran for informing us about their concurrent study[54]. E.K. acknowledges discussions with Frank Schindler regarding their related work[55]. A.V. and E.K. were funded by a Simons Investigator grant (AV) and by the Simons Collaboration on Ultra-Quantum Matter, which is a grant from the Simons Foundation (618615, A.V.).

## Author contributions

E.K. and A.V. contributed equally to the conception, execution, and writing of the project.

## Competing interests

The authors declare no competing interests.
