## [Peer Review File · Nature Communications]

Baby skyrmions in Chern ferromagnets: A topological mechanism for spin-polaron formation in twisted bilayer grapheneREVIEWER COMMENTS

Reviewer #1 (Remarks to the Author):

In their manuscript "From electrons to baby skyrmions in Chern ferromagnets: A topological mechanism for spin-polaron formation in twisted bilayer graphene" the authors address the formation of electron + spin-flip excitations in Chern ferromagnets. The authors argue that this problem is related to - and in a sense the opposite limit of - the problem of smooth topological spin textures in quantum hall ferromagnets. Whereas in the case of the latter a real space description (of the textures) is natural, this is ill-suited for the former. The authors thus choose a momentum-space approach and hence address an old and well-known problem from a new perspective. This is inspired by the recent developments in twisted bilayer graphene (TBG), where one prominent approach to addressing the physics relies on the hidden Landau level structure of the moire bands. The authors also take this approach (which departs from the so-called chiral limit) and interpret their general results in light of the particulars of TBG.

The authors focus mainly on establishing the binding energy of spin-polaron formation as compared to the energy of single-particle electron excitations. They show that in the flat dispersion limit the spin-polaron always have lower energy and they trace the attraction between electrons and spin-flips back to the topology of the band. They further show that the binding energy depends on the bandwidth, with a threshold value beyond which electron excitations have lower energy. The authors clearly state the caveats and assumptions of their work, the strengthens the interpretation and significance of their work.

I think this work is an important contribution which strikes a remarkable balance between generality and practical relevance for a currently intensely studied material class. The authors in some sense reconsider a classic problem (ie quantum Hall ferromagnets) from a new angle and with an urgent experimental motivation, adding significant understanding. I recommend publication but do have a number questions/comments which should be addressed first.

(1) I thought that equation 7 is a little bit confusing. It is not immediately obvious whether B should be interpreted as a real magnetic field or as a momentum space Berry curvature. For a magnetic field one would expect the magnetic length to scale inversely with the magnetic flux density. Furthermore, taking the first equality and integrating over the BZ (I am assuming A_{BZ} is the area of the Brillouin zone) would seem to give (essentially) the Chern number, which suggests B is the constant Berry curvature required to obtain C . The subsequent discussion suggests B and $B(r)$ is real magnetic field, however.

(2) A second question concerns the flatness of the dispersion. If I understand correctly the bare dispersion ϵ_0 is zero in the perfect chiral limit. Hence do the authors simply set ϵ_F to zero as well? When $\kappa \neq 0$ I believe the bare bands acquire dispersion, thus do the authors simply assume ϵ_F cancels that? Does this lead to inconsistencies if neither V_q or $\lambda_q(k)$ can be assumed zero? In Eq 11, isn't the dispersion still assumed to be flat, and therefore couldn't one just ignore $\epsilon(k_0+q)$?

(3) Minor question/point: in Eq 10, should ψ be ϕ ? Or otherwise how is ψ defined? Immediately following Eq 10, when the authors speak of "This operator", do they mean the operator which has matrix elements given by Eq 10?

(4) A more broad question: the fact that the authors find that the attraction is intrinsically related to the topology of a Chern band, is that essentially the "mirror image" of the statement that skyrmions (or more broadly real space topological spin textures) in quantum Hall ferromagnets carry charge? If this is a correct picture, meaning that this problem is indeed in a sense the opposite limit of quantum Hall ferromagnet problem, could there be other consequences? The effective action of quantum Hall ferromagnets can have a Hopf term, determining the spin and statistics of a skyrmions -- is there hope of an analog here?

Reviewer #2 (Remarks to the Author):

The authors present an analysis of the energetics of a small spin polaron (electron/hole + 1 spin flip) in twisted bilayer graphene near the magic angle, near integer fillings. They make use of the fact that the Hilbert space for small polarons is small enough that they can diagonalize the Hamiltonian projected to this subspace of the many-body Hilbert space. This approach has two aspects which are complementary to conventional Hartree-Fock calculations of skyrmions in quantum Hall ferromagnets and their Chern band cousins: (i) This analysis proceeds in momentum space, and it is evident where the band topology enters. (ii) Even when the polarons are not the lowest energy excitations they may appear as resonances, and alter the dynamics at low doping near an integer filling.

The work is interesting, well-done, and well-presented. Enough technical details are provided for the work to be reproduced by interested parties.

I have a few things I was puzzled by, and wonder if the authors are able to clarify.

1. The nonlinear sigma model approach indicates that in the chiral limit ($w_{AA}=0$), when the dispersion is flat, there are no anisotropy terms, so the skyrmions are of infinite size to minimize the Coulomb interaction. The energy dispersion, driven by hopping between different Chern sectors, and residual interactions give rise to anisotropies which make the skyrmions finite. One would expect that as the anisotropies increase, the size of the lowest energy skyrmion decreases in steps, until it becomes simply an electron or hole. Thus, it would seem that the range of anisotropies for which the small polaron studied by the authors is the lowest excitation would be very small. Is this expectation correct? Is there some experimental way of tuning the doping or some other parameter and observe a regime dominated by small polaron physics?

2. On page 6, the authors remark that the phase boundary where the small polaron becomes unstable occurs for a particular ratio of m_{eff}/m_e , where m_e is the bare electron mass. How and where does the bare electron mass enter the Hamiltonian? Is this simply a phenomenological observation, or does it have some deeper significance?

3. As in any approximation, it would be useful to estimate the effects of the parts left out of the approximation. The authors considered the subspace of a charge + one spin-flip. The full Hamiltonian connects this subspace to that of a charge + 2 spin-flips etc etc. Is it possible to make an estimate (perhaps even qualitatively) of the size and physical effects of these neglected parts of the Hamiltonian?

We thank the referees for their careful reading of the manuscript, their insightful comments and their positive assessment of our work. Below we provide a point-by-point reply to the referee comments and the corresponding changes in the manuscript which are highlighted in blue in the resubmitted manuscript.

Sincerely,

Eslam Khalaf and Ashvin Vishwanath

Reviewer #1 (Remarks to the Author):

In their manuscript "From electrons to baby skyrmions in Chern ferromagnets: A topological mechanism for spin-polaron formation in twisted bilayer graphene" the authors address the formation of electron + spin-flip excitations in Chern ferromagnets. The authors argue that this problem is related to - and in a sense the opposite limit of - the problem of smooth topological spin textures in quantum hall ferromagnets. Whereas in the case of the latter a real space description (of the textures) is natural, this is ill-suited for the former. The authors thus choose a momentum-space approach and hence address an old and well-known problem from a new perspective. This is inspired by the recent developments in twisted bilayer graphene (TBG), where one prominent approach to addressing the physics relies on the hidden Landau level structure of the moire bands. The authors also take this approach (which departs from the so-called chiral limit) and interpret their general results in light of the particulars of TBG

The authors focus mainly on establishing the binding energy of spin-polaron formation as compared to the energy of single-particle electron excitations. They show that in the flat dispersion limit the spin-polaron always have lower energy and they trace the attraction between electrons and spin-flips back to the topology of the band. They further show that the binding energy depends on the bandwidth, with a threshold value beyond which electron excitations have lower energy. The authors clearly state the caveats and assumptions of their work, the strenghtens the interpretation and significance of their work.

I think this work is an important contribution which strikes a remarkable balance between generality and practical relevance for a currently intensely studied material class. The authors in some sense reconsider a classic problem (ie quantum Hall ferromagnets) from a new angle and with an urgent experimental motivation, adding significant understanding. I recommend publication but do have a number questions/comments which should be addressed first.

We thank the referee for the careful reading and positive assessment of our work.

(1) I thought that equation 7 is a little bit confusing. It is not immediately obvious whether B should be interpreted as a real magnetic field or as a momentum space Berry curvature. For a magnetic field one would expect the magnetic length to scale inversely with the magnetic flux density. Furthermore, taking the first equality and integrating over the BZ (I am assuming A_{BZ} is the area of the Brillouin zone) would

seem to give (essentially) the Chern number, which suggests B is the constant Berry curvature required to obtain C . The subsequent discussion suggests B and $B(r)$ is real magnetic field, however.

We thank the referee for pointing out this issue. Indeed, we have used an inconsistent notation where we used the same variable B to represent both the real magnetic field and the momentum space magnetic field (i.e. average Berry curvature). We have fixed this in the updated manuscript by distinguishing the two using B for real field and calligraphic B for momentum space field.

(2) *A second question concerns the flatness of the dispersion. If I understand correctly the bare dispersion ϵ_0 is zero in the perfect chiral limit. Hence do the authors simply set ϵ_F to zero as well? When $\kappa \neq 0$ I believe the bare bands acquire dispersion, thus do the authors simply assume ϵ_F cancels that? Does this lead to inconsistencies if neither V_q or $\lambda_q(k)$ can be assumed zero? In Eq 11, isn't the dispersion still assumed to be flat, and therefore couldn't one just ignore $\epsilon(k_0+q)$?*

It is correct that in the chiral limit, ϵ_0 is zero. However, since there are several possible terms which can add to this dispersion including projecting out the remote bands and Hartree corrections from the filled bands, we prefer to take ϵ_0 as an independent parameter of the model. The total quasiparticle dispersion is given by the sum of ϵ_0 and the interaction generated dispersion (which as the referee pointed is non-zero if the form-factor $\lambda_q(k)$ has a non-trivial k dependence). What we call the flat quasiparticle dispersion is the limit when this total dispersion vanishes. If we take the chiral limit with no contribution from the remote bands, then this limit is equivalent to $\epsilon_F = 0$ which is generally not true. However, an excellent approximation to this limit, is the hole (electron) dispersion at $\nu=+1$ ($\nu=-1$) where the Hartree contribution to ϵ_0 almost perfectly cancels the Fock dispersion ϵ_F as seen in Fig. 4. We have added a clarification to this point in the updated manuscript.

(3) *Minor question/point: in Eq 10, should ψ be ϕ ? Or otherwise how is ψ defined? Immediately following Eq 10, when the authors speak of "This operator", do they mean the operator which has matrix elements given by Eq 10?*

We believe the referee is referring to the arxiv version. It is correct that there was a typo in Eq. 10 where ψ is written instead of ϕ . This was fixed in the version which was submitted to the journal and moved to the methods section (Eq. 24). It is also correct that "This operator" refers to the operator whose matrix elements are given by the equation above. We have added a small clarification to this point in the updated manuscript.

(4) *A more broad question: the fact that the authors find that the attraction is intrinsically related to the topology of a Chern band, is that essentially the "mirror image" of the statement that skyrmions (or more broadly real space topological spin textures) in quantum Hall ferromagnets carry charge? If this is a correct picture, meaning that this problem is indeed in a sense the opposite limit of quantum Hall ferromagnet problem, could there be other consequences? The effective action of quantum Hall ferromagnets can have a Hopf term, determining the spin and statistics of a skyrmions -- is there hope of an analog here?*

For the question about statistics, we believe this is something which is manifest in our formulation but subtle in the effective field theory. In our formulation, the excitation is manifestly a fermion since it consists of two particles plus one hole (or a particle + spin flip). In contrast, in the field theory, a Hopf term is needed to ensure that the skyrmion, which is a topological texture of a bosonic field, is fermionic.

Regarding the possibility of novel consequences of our picture, we believe that our perspective can shed lights on aspects of charge excitations that are not easy to see from the field theory perspective. For instance, in a band with Chern number $C > 1$, the field theory skyrmions have a charge that is a multiple of $C e$, which correspond to a bound state of $|C|$ electrons to one or more spin flips, but our theory allows in principle for bound state of any charge. We plan to investigate these questions systematically in future works.

Reviewer #2 (Remarks to the Author):

The authors present an analysis of the energetics of a small spin polaron (electron/hole + 1 spin flip) in twisted bilayer graphene near the magic angle, near integer fillings. They make use of the fact that the Hilbert space for small polarons is small enough that they can diagonalize the Hamiltonian projected to this subspace of the many-body Hilbert space. This approach has two aspects which are complementary to conventional Hartree-Fock calculations of skyrmions in quantum Hall ferromagnets and their Chern band cousins: (i) This analysis proceeds in momentum space, and it is evident where the band topology enters. (ii) Even when the polarons are not the lowest energy excitations they may appear as resonances, and alter the dynamics at low doping near an integer filling.

The work is interesting, well-done, and well-presented. Enough technical details are provided for the work to be reproduced by interested parties.

We thank the referee for the careful reading and positive assessment of our work

I have a few things I was puzzled by, and wonder if the authors are able to clarify.

1. The nonlinear sigma model approach indicates that in the chiral limit ($w_{AA}=0$), when the dispersion is flat, there are no anisotropy terms, so the skyrmions are of infinite size to minimize the Coulomb interaction. The energy dispersion, driven by hopping between different Chern sectors, and residual interactions give rise to anisotropies which make the skyrmions finite. One would expect that as the anisotropies increase, the size of the lowest energy skyrmion decreases in steps, until it becomes simply an electron or hole. Thus, it would seem that the range of anisotropies for which the small polaron studied by the authors is the lowest excitation would be very small. Is this expectation correct? Is there some experimental way of tuning the doping or some other parameter and observe a regime dominated by small polaron physics?

First, we would like to highlight that our main result is the existence of a parameter regime where the electron is not the lowest energy charge e excitation. This does not rule out the possibility of larger polarons with multiple spin flips being even more favored or that these are favored in the regime where we find the electron to be more favored than the small polaron. However qualitatively once we are in

the domain of stability of polarons, we expect significant modification of the physics compared to single electrons, which is why this calculation is important even if it cannot determine precisely the number of bound spin flips. In other words, we believe that once spin polarons are formed, no matter the precise size, their properties are well approximated by the simplest nontrivial one, the single spin flip bound states studied here. For instance, they will be very heavy particles with very flat dispersion similar to the small polaron studied here.

As for the effect of different anisotropies, it is true that in the isotropic limit, we expect large polarons with multiple spin flips to be even more favored than the ones with single spin flip considered here. However, it is generally possible to realize an experimental regime where the small polaron is favored over larger polarons/skyrmions but also over electrons by applying a Zeeman field. It is known from the quantum Hall context that the skyrmion size shrinks monotonically with increasing Zeeman field. This means that there is a range of fields for which the polaron considered here is the lowest energy excitation provided it is already lower in energy than the electron to begin with. When applying this to context of twisted bilayer graphene (TBG), there is a few caveats to consider. First, skyrmions in TBG could form in the spin, pseudospin (valley) degrees of freedom, or a combination of them. The anisotropies that act as Zeeman field for these different cases are different. For spin skyrmions, this is the actual Zeeman field which can be realized by applying an in-plane magnetic field (assuming orbital coupling is neglected). For pseudospin skyrmion, a sublattice potential generated for instance from alignment with the hBN substrate will realize an effective Zeeman field. We also note that terms which couple the Chern sectors that we have neglected in our analysis act as an effective Zeeman field and can favor the small polarons we consider over larger ones. We have added a discussion explaining the effect of Zeeman field and coupling between Chern sectors on the polaron energy in the updated manuscript.

2. *On page 6, the authors remark that the phase boundary where the small polaron becomes unstable occurs for a particular ratio of m_{eff}/m_e , where m_e is the bare electron mass. How and where does the bare electron mass enter the Hamiltonian? Is this simply a phenomenological observation, or does it have some deeper significance?*

The bare mass is simply used to obtain a dimensionless ratio that can be compared directly to experimentally extracted values (see for example Fig. 5e in Ref. 1 or Fig. 3b in Ref. 2). The observation of a particular value of the ratio for which the polaron becomes unstable is a phenomenological observation. The main implication is that the polaron stability seems mostly sensitive to the effective mass, i.e. the curvature at the band minimum, rather than the total bandwidth which can become very large without destabilizing the polaron. This can be seen in Fig. 3d where the polaron remains stable despite the substantial bandwidth. The main reason for this is that the polaron can evade regions of large band dispersion provided they are away from the band minimum as shown in Fig. 2c.

3. *As in any approximation, it would be useful to estimate the effects of the parts left out of the approximation. The authors considered the subspace of a charge + one spin-flip. The full Hamiltonian connects this subspace to that of a charge + 2 spin-flips etc etc. Is it possible to make an estimate (perhaps even qualitatively) of the size and physical effects of these neglected parts of the Hamiltonian?*

The two terms that couple the two sectors come from the dispersion and the deviation from the chiral limit. These can be estimated from the effective field theory (Refs. [20] and [21]). For charge e polarons in a single sector, these terms induce an effective Zeeman coupling from the spin/pseudospins in the other sector. We have added a brief discussion estimating these effects to the text.

REVIEWERS' COMMENTS

Reviewer #2 (Remarks to the Author):

I am happy with the response of the authors to my questions, and the changes they have made to the manuscript. It is definitely suitable for publication in Nature Communications.